# Surveillance for face mask compliance, Chennai, Tamil Nadu, India, October-December, 2020

**Jagadeesan M.**[1], **Polani Rubeshkumar**[2], **Mohankumar Raju**[2],
**Manikandanesan Sakthivel**[2], **Sharan Murali**[2], **Ramya Nagarajan**[2],
**Muthappan Sendhilkumar**[2], **Irene Sambath**[2], **Kumaravel Ilangovan**[2],
**Dineshkumar Harikrishnan**[2], **Vettrichelvan Venkatasamy**[2], **Parasuraman Ganeshkumar**[3],
**Madhusudhan Reddy**[1], **Prabhdeep Kaur**[2]*

1 Public Health Department, Greater Chennai Corporation, Chennai, Tamil Nadu, India, 2 Division of Noncommunicable Diseases, ICMR-National Institute of Epidemiology, Chennai, Tamil Nadu, India, 3 Division of Epidemiology, ICMR-National Institute of Epidemiology, Chennai, Tamil Nadu, India

* kprabhdeep@gmail.com

**Data Availability Statement:** The datasets generated and analysed during the current study have been deposited in the Mendeley repository at https://data.mendeley.com/datasets/b2yzdfv9rt/1 (DOI: 10.17632/b2yzdfv9rt.1).

## Abstract

### Purpose

Government of Tamil Nadu, India, mandated the face mask wearing in public places as one of the mitigation measures of COVID-19. We established a surveillance system for monitoring the face mask usage. This study aimed to estimate the proportion of the population who wear face masks appropriately (covering nose, mouth, and chin) in the slums and non-slums of Chennai at different time points.

### Methods

We conducted cross-sectional surveys among the residents of Chennai at two-time points of October and December 2020. The sample size for outdoor mask compliance for the first and second rounds of the survey was 1800 and 1600, respectively, for each of the two sub-groups–slums and non-slums. In the second round, we included 640 individuals each in the slums and non-slums indoor public places and 1650 individuals in eleven shopping malls. We calculated the proportions and 95% confidence interval (95%CI) for the mask compliance outdoors and indoors by age, gender, region, and setting (slum and non-slum).

### Results

We observed 3600 and 3200 individuals in the first and second surveys, respectively, for outdoor mask compliance. In both rounds, the prevalence of appropriate mask use outdoors was significantly lower in the slums (28%-29%) than non-slum areas (36%-35%) of Chennai (p<0.01). Outdoor mask compliance was similar within slum and non-slum subgroups across the two surveys. Lack of mask use was higher in the non-slums in the second round (50%) than in the first round of the survey (43%) (p<0.05). In the indoor settings in the 2nd

**Funding:** This study was supported by the Intramural fund of ICMR-National Institute of Epidemiology, Chennai, India. The funders had no role in study design, data collection, and analysis, decision to publish, or preparation of the manuscript.

**Competing interests:** The authors have declared that no competing interests exist.

survey, 10%-11% among 1280 individuals wore masks appropriately. Of the 1650 observed in the malls, 947 (57%) wore masks appropriately.

## Conclusion

Nearly one-third of residents of Chennai, India, correctly wore masks in public places. We recommend periodic surveys, enforcement of mask compliance in public places, and mass media campaigns to promote appropriate mask use.

## Introduction

Wearing a mask is one of the simplest ways to reduce the spread of COVID-19. The World Health Organization [1] (WHO), the U.S. Centers for Disease Control and Prevention [2] (CDC), Government of India [3], and numerous other government and public health agencies have recommended using masks in public settings as one of the mitigation strategies along with physical distancing and hand washing. COVID-19 transmit from person-to-person through direct contact, contaminated fomites, and airborne transmission. Masks can prevent the spread of COVID-19 in two ways: by preventing a healthy person from acquiring the disease and by preventing an infected person from spreading the disease. Besides, the mask also decreases the hand-face contacts, which prevents the infection through contaminated fomites [4]. The consistent practice of mask use in public places irrespective of the symptoms could control the spread of COVID-19 effectively [5]. Although the benefits of mask use are known, compliance to mask use is not routinely monitored in low resource settings as a step towards mitigating the spread of COVID-19. Data from High-income Countries suggested that surveillance for mask use is useful in designing interventions and improving compliance. Mask mandate in public health places led to decreases in COVID-19 cases and mortality in a different province of the United States [5].

## Chennai, Tamil Nadu, India: The context

Chennai is the fourth-largest metropolitan city in India and is also the capital of Southern State Tamil Nadu. Spread over an area of 426 sq. km, the population of Chennai was projected to be nearly 8 million for the year 2020. Greater Chennai Corporation (GCC) is the governing body of Chennai and is administratively divided into three regions, namely North, Central, and South region. Each region is further divided into 5 Zones.

Chennai recorded 185,573 COVID-19 cases and 3,452 deaths between March 17 and October 15, 2020. On September 4, 2020, the Government of Tamil Nadu issued an order that mandates masks wearing in public places. It promulgated the existing Tamil Nadu Public Health Act that made violators be punished with a fine [5]. As lockdown and restrictions were gradually relaxed over the past few months, high mask compliance remained one of the most important interventions to prevent an upsurge of cases.

Although transmission risk is higher in indoor settings [6], the mask mandate was monitored, and authorized officials imposed fines to non-compliant individuals, predominantly in public places such as traffic signals and streets [5, 7]. We wanted to initially assess the feasibility of surveillance in easily accessible open places and subsequently expand the surveillance to indoor settings. Therefore, we selected outdoor public places for the first survey and added indoor settings in the second survey. The objective was to establish a surveillance system to

monitor the trend of outdoor mask compliance and to estimate the proportion of the population who wore a face mask appropriately at different time points in the slums and non-slums of Chennai, Tamil Nadu, India. We also estimated compliance to mask use in the indoor public places and malls in the second survey.

## Methods

We conducted two rounds of a cross-sectional survey among residents of Chennai during October 16–19, 2020, and December 26–29, 2020. We used direct in-person observation by trained observers, one of the recommended methodologies for monitoring compliance to safe behaviors [8, 9]. The geographical spread of the study sites for both the phases in given in the figure (Fig 1).

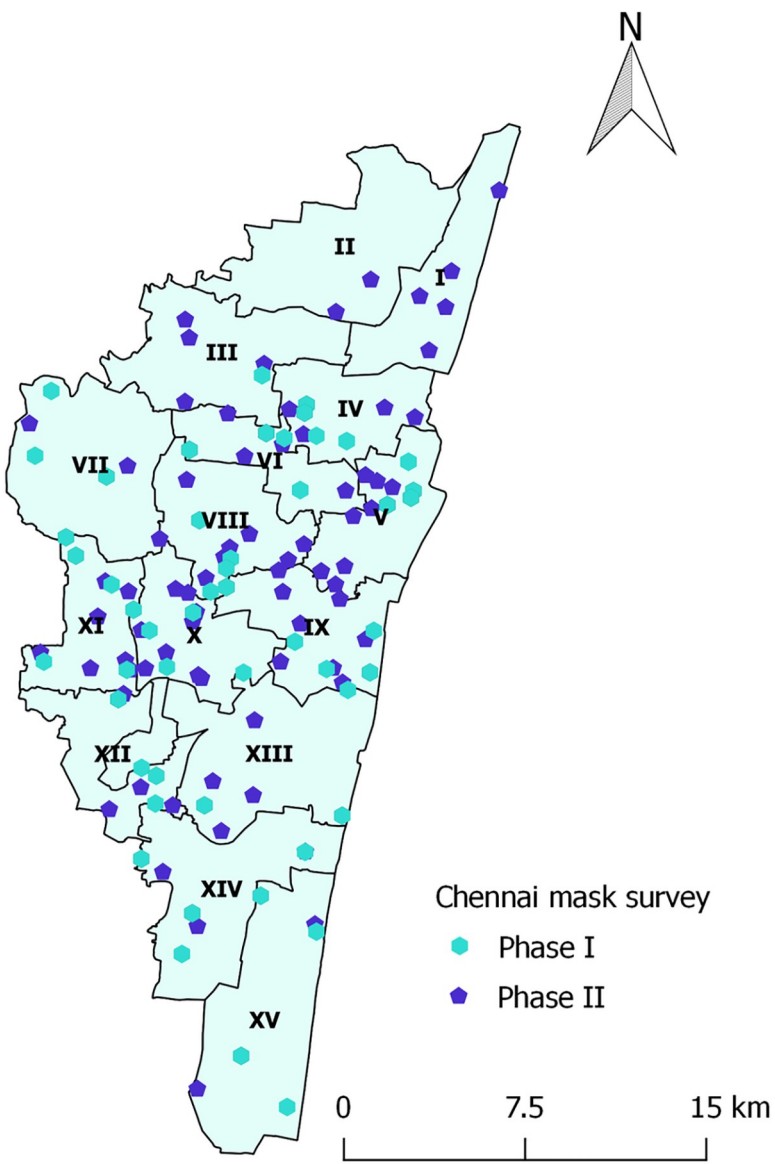

**Fig 1. Study sites of mask survey, Chennai, India, October-December 2020.**

## Operational definitions

We defined "mask" as any cloth mask, medical masks, or N95 respirators worn over the face. We categorized the mask usage into three categories: 1. Appropriate use: Mask covering the nose, mouth, and chin, 2. Inappropriate use: Mask worn either below nose or mouth, and 3. No mask: Did not wear a mask. For ease of observation and recording, we grouped the individuals into four age groups: Children/adolescents, young adults, middle-aged and older. Indoor public places included settings which were open for public and did not have any entry restrictions (e.g., grocery shop, vegetable shop, pharmacy, religious places, apparel stores), and outdoor public places were streets in the residential or commercial areas and bus stations.

## The first round of the survey

The first round of the cross-sectional survey was conducted between October 16 and October 19, 2020. During the period, the reported cases were 1140 and Test Positivity Rate (TPR) was 8.5% (Fig 2). We calculated the required sample size as 1728 (rounded to 1800) by assuming 50% mask compliance, 5% absolute precision, 95% confidence level, and 4.5 design effect [9]. We stratified all the streets in the city as slums and non-slum. We did a simple random sampling and selected two streets each from the slum (15x2 = 30) and non-slum (15x2 = 30) areas of all 15 Zones. In each of the 60 chosen streets, trained data collectors directly observed 30 consecutive individuals at two points: morning (8 to 10 AM) and evening (4 to 6 PM).

## The second round of the survey

The second-round survey was conducted between December 26 and 29, 2020. The reported cases and TPR was 295 and 2.7% respectively (Fig 2). Based on the first-round experience, we decided to include indoor public places and the shopping malls of Chennai in the study. We recalculated the required sample size as 1592 (rounded to 1600) based on 36% mask compliance (based on results from the first-round survey), 5% absolute precision, 95% confidence level, and 4.5 design effect. We randomly selected 32 streets from each of the two strata–Slum and Non-Slum. We achieved the sample size by directly observing 50 individuals in each selected street (50*32 = 1600).

We did not have any data available regarding indoor mask compliance. Therefore, we did a pilot study by observing 20 consecutive individuals in the indoor public places in already

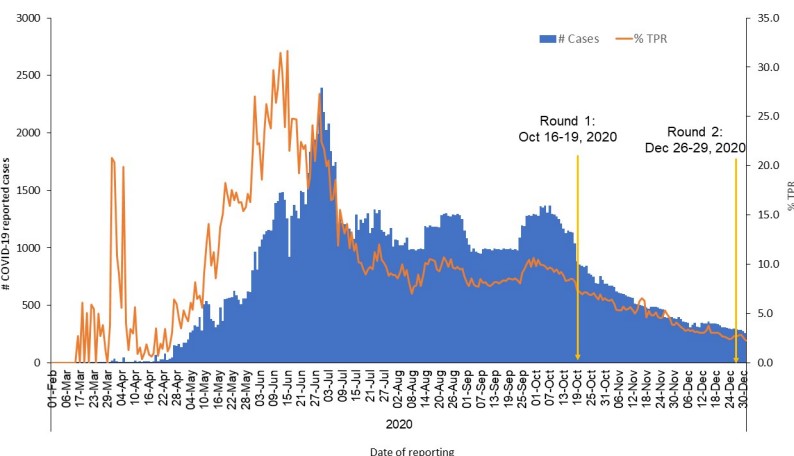

**Fig 2. COVID-19 cases, test positivity rate (%), & date of surveys, Chennai, India, October-December 2020.**

selected streets. In addition, we directly observed the mask compliance in all the 11 malls of Chennai. We used the sample size calculated for outdoors for malls. Data collectors observed 150 (rounded) individuals in each of the 11 malls to achieve a sample size of 1600.

## Data collection

We trained the data collectors on operational definitions, methodology, conducted simulation sessions to minimize the inter-observer variation. We identified the location shown by the Google map as the study site. We observed the individuals crossing from the right side to avoid potential duplication. We included pedestrians, motorcycle riders, bicycle riders, and individuals traveling in the autorickshaw and bus. We excluded the individuals traveling in a car. Inside the malls, data collectors observed the individuals on the first floor near the escalator in one direction and avoided the mall entrance and the food court. The team observed the participants from a distance to avoid the Hawthorne effect. Observers recorded 1) age group, 2) Gender, 3) whether mask worn appropriately/ inappropriately/ no mask. 4)Type of mask-Medical /cloth/ N95 mask (if any). Data collectors observed the approximate age group and gender of the individual by observation. Data were collected using Open Data Kit (ODK), an android-based mobile application. Observers collected data using their mobile phones, and the data were stored in a secured Institutional server.

## Data analysis

We analyzed the data using Epi Info version 7.2 [10]. Sociodemographic variables were summarised first as the frequency with proportions. We estimated the mask use in three categories: appropriate mask use, inappropriate mask use, and no mask. We computed the proportion and 95%CI for mask compliance by age group, gender, and regions (North, Central, and South). In each round of the survey, we compared the proportion of mask compliance by age group, gender, and region within each stratum- slums and non-slums using the chi-square test. We also compared the proportion of mask compliance by gender and the regions between the slums and non-slums in each round using the chi-square test. The difference in the proportions of mask use within the strata (slums and non-slum) between the two rounds was analyzed using the chi-square for trend test. A p-value less than 0.05 was considered significant.

## Ethical approval

The study has been approved by the Institutional Human Ethics Committee, ICMR-National Institute of Epidemiology, Chennai, India. An informed consent was not obtained because the survey team did not interview any human participants. The face mask use was observed in public places and no identifier data was collected.

## Results

### Mask use in the first-round survey

We observed 3,600 individuals, which included 1,800 individuals each in the slum and non-slum areas of Chennai. Among the 3600, 1694 (47%) were middle-aged, and nearly three-fourths (77%) were males. The study locations included residential areas, markets, bus stops, traffic signals, and religious places. The prevalence of appropriate mask use was significantly lower in the slum [28%, 95% CI: 23%-33%] than in non-slum areas of Chennai [36%, 95% CI: 31%-41%, p<0·01] (Table 1). The observed design-effect was 4.5. The appropriate mask use did not differ between males and females in the slums (27% vs. 28%, p>0.05) and in non-slums (36% vs. 36%, p>0.05) (Table 1).

**Table 1. Outdoor mask compliance in public places in the slum and non-slum areas by age group, gender, and region, Greater Chennai Corporation, Tamil Nadu, India, October & December 2020.**

| Characteristics | | October 2020 (Round 1) | | | | | | December 2020 (Round 2) | | | | | |
|---|---|---|---|---|---|---|---|---|---|---|---|---|---|
| | | Slum (N = 1800) | | | Non-Slum (N = 1800) | | | Slum (N = 1600) | | | Non-Slum (N = 1600) | | |
| | | n (%) | 95% CI | p-value | n (%) | 95% CI | p-value | n (%) | 95% CI | p-value | n (%) | 95% CI | p-value |
| Age (Years) | Children/ adolescent | 27 (18) | 11–27 | <0.001 | 42 (38) | 28–49 | <0.001 | 26 (17) | 11–26 | <0.001 | 32 (22) | 15–31 | <0.001 |
| | Young adult | 157 (31) | 25–37 | | 220 (37) | 30–44 | | 144 (30) | 24–38 | | 201 (36) | 30–42 | |
| | Middle-aged | 234 (27) | 22–33 | | 299 (36) | 31–41 | | 231 (32) | 27–39 | | 262 (38) | 33–43 | |
| | Older | 79 (28) | 22–34 | | 82 (33) | 25–42 | | 59 (23) | 16–32 | | 66 (32) | 25–30 | |
| Sex | Male | 367 (27) | 23–32 | 0.4 | 506 (36) | 31–40 | 0.9 | 342 (29) | 24–34 | 0.8 | 432 (35) | 31–40 | 0.3 |
| | Female | 130 (28) | 22–37 | | 137 (36) | 29–44 | | 118 (29) | 22–38 | | 129 (35) | 29–42 | |
| Region | North | 164 (27) | 16–38 | 0.7 | 175 (29) | 25–34 | <0.001 | 175 (25) | 17–34 | 0.001 | 115 (33) | 26–41 | 0.07 |
| | Central | 165 (28) | 19–35 | | 214 (36) | 27–46 | | 237 (34) | 26–43 | | 209 (35) | 29–46 | |
| | South | 168 (28) | 20–36 | | 254 (42) | 34–51 | | 48 (24) | 6–60 | | 237 (36) | 28–46 | |
| Overall | | 497 (28) | 23–33 | | 643 (36) | 31–41 | | 460 (29) | 24–35 | | 561 (35) | 31–39 | |

Mask compliance was similar across three regions in the slums, however, in the non-slums, southern region (42%) had higher compliance than north (29%, p<0.001) and central region (36%, p = 0.03). The appropriate mask use compliance did not differ between morning and evening both in the slums (29% vs. 26%; p>0·05) and non-slums (37% vs. 35%; p>0·05) areas of Chennai. Nearly three-fourths of the children/adolescents in the slum and half of them in non-slum areas did not wear any mask (Table 1).

## Mask use in the second-round survey

We observed 3,200 individuals in the outdoor setting, including 1,600 individuals each in the slum and non-slum areas. Among the 3200, 1402 (44%) were middle-aged and nearly three-fourths (76%) were males. Of the 1600 individuals observed in the slums, 460 (29%) wore a mask appropriately, while nearly 877 (55%) did not wear a mask. Whereas in non-slum areas, 561 (35%) wore a mask appropriately, 346 (15%) individuals wore masks either below the nose or mouth, and 799 (50%) did not wear a mask. The appropriate mask use was higher in the non-slum areas [35%, 95%CI: 31%-39%] than the slums [29%, 95%CI: 24%-35%, p<0.001] (Table 1). The mask compliance did not vary by gender (male/female) in both slums and non-slums (Table 1). Seventy-eight percent of the children/adolescents of the slums did not wear a mask. Among the 1524 individuals who wore masks in the slums and non-slums, 1220 (80%) worn cloth masks, 278 (18%) worn medical masks, and 26 (2%) used N95 masks.

We observed 1280 individuals in indoor public places such as pharmacies, departmental stores, vegetable shops, apparel stores, and religious sites in slums and non-slums. The majority (80%) of the indoor participants were young adults and middle-aged, and three-fourths were males. In the slums, 72 (11%) wore masks appropriately, 134 (20%) wore masks either below the nose or mouth, while 432 (68%) did not wear a mask at all. Similarly, only 67 (10%) wore masks appropriately in the non-slum areas, 141 (22%) wore masks inappropriately, and 432 (68%) did not wear a mask at all (Table 2). The mask compliance was higher among males than females of slums (12% vs. 7%; p<0.05), and no difference in mask compliance by gender in the non-slums (12% vs. 6%; p>0.05) (Table 2). Of the total 414 individuals who wore masks indoors of slum and non-slum, four out of five wore cloth masks, 66 (16%) wore medical masks, and 3 (1%) used N95 masks.

On comparing the appropriate mask use indoors and outdoors, In slum areas, indoor places had lower compliance than outdoors (11% vs 29%, p<0.001). Similarly, in the non-slums, the

**Table 2. Indoor mask compliance in public places in the slum and non-slum areas by age group, gender, and region, Greater Chennai Corporation, Tamil Nadu, India, December 2020.**

| Characteristics | | Slum (N = 640) | | Non-slum (N = 640) | |
|---|---|---|---|---|---|
| | | n (%) | p-value | n (%) | p-value |
| Age (Years) | Children/ adolescent | 1 (3) | 0.009 | 2 (8) | 0.01 |
| | Young adult | 18 (13) | | 17 (10) | |
| | Middle-aged | 38 (14) | | 37 (10) | |
| | Older | 15 (17) | | 11 (13) | |
| Sex | Male | 61 (12) | 0.03 | 57 (12) | 0.2 |
| | Female | 11 (7) | | 10 (6) | |
| Region | North | 25 (9) | 0.003 | 7 (5) | <0.001 |
| | Central | 41 (15) | | 39 (16) | |
| | South | 6 (8) | | 21 (8) | |
| Overall | | 72 (11) | | 67 (10) | |

indoor mask compliance was significantly lower than the outdoor public places (10% vs 35%, p<0.001).

Among the 1650 individuals observed across the 11 malls in Chennai, one-in-two were young adults, and 1036 (63%) were males. Nearly half (57%) wore masks appropriately, 367 (22%) wore inappropriately, and 336 (21%) did not wear a mask in the malls of Chennai. Among the 1314 individuals who wore masks in the malls, 894 (68%) used cloth masks, 318 (24%) used medical masks, and 102 (8%) used N95 masks.

## Outdoor mask compliance comparison of two surveys

The appropriate mask use did not differ within the slums (28% vs. 29%, p>0.05) and non-slums (36% vs. 35%, p>0.05) (Table 3) subgroups of Chennai between the two rounds of the survey. There was no significant difference in the appropriate mask use between the two surveys within gender by slum and non-slum (Table 3). The appropriate mask use is reduced among the children/adolescents of non-slums in the second round than the first round (22% vs. 38%; p<0.05) (Table 3). Mask compliance is increased among the middle-aged population of slums in the second round than the first round of survey (27% vs. 32%; p<0.05) (Table 3). Similarly, mask compliance is increased in the Slums of Central region of Chennai in the second round than the first round (34% vs. 28%; p<0.05) (Table 3).

**Table 3. Appropriate mask use and no mask use by rounds of surveys in the slum areas of Chennai, India, 2020.**

| | | Slum | | | | | | Non-Slum | | | | | |
|---|---|---|---|---|---|---|---|---|---|---|---|---|---|
| | | Appropriate mask use, n (%) | | | No mask use, n (%) | | | Appropriate mask use, n (%) | | | No mask use, n (%) | | |
| Characteristics | | Oct-20 | Dec-20 | p-value | Oct-20 | Dec-20 | p-value | Oct-20 | Dec-20 | p-value | Oct-20 | Dec-20 | p-value |
| Age (Years) | Children/ adolescent | 27 (18) | 26 (17) | 0.9 | 114 (74) | 119 (78) | 0.44 | 42 (38) | 32 (22) | 0.006 | 56 (50) | 103 (72) | <0.001 |
| | Young adult | 157 (31) | 144 (30) | 0.82 | 279 (55) | 269 (57) | 0.6 | 220 (37) | 201 (36) | 0.76 | 288 (48) | 303 (54) | 0.04 |
| | Middle-aged | 234 (27) | 231 (32) | 0.02 | 477 (56) | 338 (48) | <0.001 | 299 (36) | 262 (38) | 0.33 | 334 (40) | 287 (42) | 0.5 |
| | Older | 79 (28) | 59 (23) | 0.17 | 152 (53) | 151 (58) | 0.3 | 82 (33) | 66 (32) | 0.87 | 105 (42) | 106 (52) | 0.04 |
| Sex | Male | 367 (27) | 342 (29) | 0.46 | 758 (56) | 646 (54) | 0.23 | 506 (36) | 432 (35) | 0.73 | 617 (43) | 609 (49) | 0.002 |
| | Female | 130 (28) | 118 (29) | 0.83 | 264 (58) | 231 (57) | 0.8 | 137 (36) | 129 (35) | 0.81 | 166 (44) | 190 (53) | 0.02 |
| Region | North | 164 (27) | 175 (25) | 0.33 | 381 (64) | 407 (58) | 0.43 | 175 (29) | 115 (33) | 0.27 | 298 (50) | 193 (55) | 0.001 |
| | Central | 165 (28) | 237 (34) | 0.01 | 297 (50) | 348 (50) | 0.02 | 214 (36) | 209 (35) | 0.33 | 245 (41) | 285 (47) | 0.4 |
| | South | 168 (28) | 48 (24) | 0.26 | 344 (57) | 122 (61) | 0.48 | 254 (42) | 237 (36) | 0.17 | 240 (40) | 321 (50) | 0.003 |
| Overall | | 497 (28) | 460 (29) | 0.46 | 1022 (57) | 877 (55) | 0.24 | 643 (36) | 561 (35) | 0.68 | 783 (43) | 799 (50) | <0.001 |

The lack of mask use was higher in the non-slums in the second round than in the first round of the survey (50% vs. 43%, p<0.01). The practice of not wearing mask increased among the Children/adolescents (72% vs. 50%, p<0.001), young adults (54% vs. 48%, p<0.05), males (49% vs. 43%, p<0.01) and females (53% vs. 44%; p<0.05) in the second round compared to the first round (Table 3). The practice of not wearing mask is decreased among the middle-aged population of the slums of Chennai (56% vs. 48%; p<0.001) (Table 3).

## Discussion

Repeat surveys to monitor mask compliance at two-time points, October and December 2020, suggested low compliance with no remarkable change over time. Although the law mandated mask use and the authority impose a penalty for violators, mask compliance was low in outdoor public places. In areas with strict enforcement, such as shopping malls, the compliance was higher. Many countries witnessed the second wave of the pandemic, and few countries reported the circulation of UK and South African variants of COVID-19 [11]. Despite the circulation of newer strains and vaccine introduction, wearing masks remained the core mitigation strategy. Studies have proven the protective role of face masks in preventing COVID-19 transmission [12]. Mask mandate in the United States of America is associated with the decreased growth rate of COVID-19, hospitalization, and mortality [13–15]. Therefore, it is critical to monitor mask compliance for COVID-19 control.

In our setting, nearly one-fifth did not wear the mask appropriately. Lack of appropriate use or loosely fitted masks offered no protection to the wearer and also others. These could be minimized by wearing the mask with knotted ear loops [16]. We observed decline in the mask compliance in the southern region of the city compared to other regions. This can be possibly attributed to the low risk perception among residents of the southern region as the region reported the lowest cases throughout the first wave. The reasons for overall increase in non-compliance of the mask were multi-factorial. Between the first and second round of survey, there was decline in reported cases and TPR in the city, thus many activities such as gatherings were resumed. The relaxations of restrictions might have led to perception that pandemic was over and masks were not required. Wearing a face mask is considered as a sign of weakness and stigma in some settings which influences the intention to wear mask [17]. A study from Greece also reported the low compliance of mask usage among the younger age group and males [18]. Individuals who wear spectacles had foggy vision by wearing mask [19]. Other cities such as Honolulu, USA also reported geographic variations in mask compliance [20]. Although reasons may vary in various settings, sustaining high compliance to mask use is challenging in many cities worldwide.

We observed higher mask compliance outdoors than indoors, whereas a similar observation study conducted in the six universities of the USA reported higher indoor mask compliance than outdoors [21]. There is a system in a place to collect fines for violations of mask compliance, and the fine is Rs 500/-, which is equivalent to one day wage of an unskilled labourer. Despite these measures, people did not prefer to wear mask during the study period, may be because cases were declining [5, 7].

We established surveillance at a low-cost using a rapid methodology which was feasible in our setting. The overall expenses incurred for each round of the survey was only 70,000 (~ $1000), which included daily wages of the staff and travel. Observation-based surveys can provide rapid feedback to the public health authorities on the prevalence, behavior, and type of mask used in the population. Using this methodology, we could collect substantial amounts of data at regular intervals at low cost and review the data quickly for public health action.

Furthermore, the study would help evaluate interventions' effectiveness over time. We could reinforce the need for increasing awareness and enhancing the enforcement of mask mandates.

The major limitation of the study was the inter-observer variation in identifying the age group of the individual. We minimized the inter-observer variation by training and standardizing the protocols. However, the bias was independent of the outcome (prevalence of appropriate mask use).

## Conclusion

Nearly one-third of residents of Chennai, India, across the different age groups and gender, correctly wore masks in public places. Although overall mask compliance did not change over time, the lack of mask use increased in the non-slum areas of South Chennai over time. The mask compliance was highest in the malls of Chennai. We recommend periodic surveys, enforcement of mask compliance in public places, and mass media campaigns to promote appropriate mask use. The mask compliance surveillance should be scaled up to all major cities in India and other similar settings.

## Author Contributions

**Conceptualization:** Jagadeesan M., Polani Rubeshkumar, Mohankumar Raju, Sharan Murali, Ramya Nagarajan, Dineshkumar Harikrishnan, Parasuraman Ganeshkumar, Madhusudhan Reddy, Prabhdeep Kaur.

**Data curation:** Polani Rubeshkumar, Mohankumar Raju, Manikandanesan Sakthivel, Sharan Murali, Ramya Nagarajan, Muthappan Sendhilkumar, Irene Sambath, Kumaravel Ilangovan, Vettrichelvan Venkatasamy.

**Formal analysis:** Polani Rubeshkumar, Mohankumar Raju, Manikandanesan Sakthivel, Sharan Murali, Ramya Nagarajan, Parasuraman Ganeshkumar, Prabhdeep Kaur.

**Methodology:** Jagadeesan M., Polani Rubeshkumar, Mohankumar Raju, Manikandanesan Sakthivel, Sharan Murali, Ramya Nagarajan, Parasuraman Ganeshkumar, Madhusudhan Reddy, Prabhdeep Kaur.

**Writing – original draft:** Jagadeesan M., Polani Rubeshkumar, Prabhdeep Kaur.

**Writing – review & editing:** Jagadeesan M., Polani Rubeshkumar, Mohankumar Raju, Manikandanesan Sakthivel, Sharan Murali, Ramya Nagarajan, Muthappan Sendhilkumar, Irene Sambath, Kumaravel Ilangovan, Dineshkumar Harikrishnan, Vettrichelvan Venkatasamy, Parasuraman Ganeshkumar, Madhusudhan Reddy, Prabhdeep Kaur.

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
