## [Decision Letter · Decision Letter 0]

3 May 2021

PONE-D-21-08966

Surveillance for face mask compliance, Chennai, Tamil Nadu, India, October-December, 2020

PLOS ONE

Dear Dr. Kaur,

Thank you for submitting your manuscript to PLOS ONE. After careful consideration, we feel that it has merit but does not fully meet PLOS ONE’s publication criteria as it currently stands. Therefore, we invite you to submit a revised version of the manuscript that addresses the points raised during the review process.

We look forward to receiving your revised manuscript.

Kind regards,

Amitava Mukherjee, ME, Ph.D.

Academic Editor

PLOS ONE

Journal Requirements:

"This study was funded by the Intra-mural fund of ICMR- National Institute of

Epidemiology, Chennai, India. The funders had no role in study design, data collection,

and analysis, decision to publish, or preparation of the manuscript."

Reviewers' comments:

Reviewer's Responses to Questions

**Comments to the Author**

1. Is the manuscript technically sound, and do the data support the conclusions?

Reviewer #1: Partly

Reviewer #2: Partly

2. Has the statistical analysis been performed appropriately and rigorously? 

Reviewer #1: No

Reviewer #2: Yes

3. Have the authors made all data underlying the findings in their manuscript fully available?

Reviewer #1: No

Reviewer #2: Yes

4. Is the manuscript presented in an intelligible fashion and written in standard English?

Reviewer #1: Yes

Reviewer #2: Yes

5. Review Comments to the Author

Reviewer #1: Referee Report

This paper reports the result of a surveillance study on masking prevalence in Chennai, Tamil Nadu, India. The authors carried out two surveillance surveys twice in Chennai in October and December 2020. The sample was chosen such that it included both slum and non-slum areas. Further, the authors conducted an additional surveillance exercise in indoor spots close to the outdoor surveillance spots.

Comments:

1) From the abstract or the introduction, this study's primary purpose is not very clear. It would be helpful for authors to add a clear statement of purpose to the paper for ease of reading.

2) The discussion in this paper will be greatly enriched if the authors clarify the policy context. For instance, as a reader, I would like to know about the policing norm around masking and the associated fines. While authors mention it in a footnote, it should be in the main text. Also, they could explain to the reader if it is a small or significant amount by comparing it to local minimum wages. It would be helpful also to have details on the existing information campaigning around masking in Chennai.

3) Authors did not explain how locations were chosen. A map with surveillance points will help the reader imagine this exercise more intuitively. It is okay if these locations were chosen, keeping the ease of observation in mind, but this, or any other underlying reasoning out to be stated.

4) The study will also be more relevant if we can say something about masking rates in the city based on the masking rates observed in the survey. At least they comment on how the chosen surveillance points differ from the rest of the city, if at all.

5) Is the design effect taken from prior literature? What about clustering? Further details about the sampling design would be helpful.

6) I thought that the comparison across time periods was the most interesting. It can be complemented by information on disease spread over time: for instance, a chart that shows the evolution of the TPR over time for Chennai and the survey periods are highlighted along with making rates

7) Some other comparisons are not very interesting (for instance, by gender). The authors also don't explain the reason for other observed differences (by region).

Other minor comments:

1) Combine Replace Tables 1 and 2 with Table 4.

2) Clarify in table notes what values in the parenthesis mean, etc.

3) Review endnotes. For instance, endnote 7 links to a media report about fine for not masking, but it is linked to part where authors are discussing survey design.

Reviewer #2: The following corrections are suggested:

-Abstract: The subsection title "Introduction" could be replaced with the title "Purpose"

-Introduction / p. 5, 2nd paragraph: Please remove the comma before the full stop in "...settings,."

-Methods: Please describe in more details the age classification. Do you estimate approximately the age of each individual according to his/her appearance?

-Results/ p. 8, last paragraph: "42% vs. 29% vs. 36%": Do you mean "42% vs. 29% for non-slums and slums, respectively" or "42% vs. 36% for non-slums and slums, respectively". Please make it clear.

-Results, (eg Tables 1, 2, etc): When p>0.001), you are suggested to write the exact p value.

-Results, p. 8: "The appropriate mask use compliance did not differ between morning and evening both in the slums (29% vs. 26%; p>0·05) and non-slums (37% vs. 35%; p>0·05) areas of Chennai": It would be very interesting if you made this comparison for the second round both for outdoor and indoor public places, too.

-Results, p. 11: "Among the 1524 individuals who wore masks in the slums and non-slums, 1220 (80%) worn cloth masks, 278 (18%) worn medical masks, and 26 (2%) used N95 masks.": It would be very interesting if you made this comparison for the first round, too.

-Additionally, the data for the type of mask (Medical /cloth/ N95 mask) could be added in the comparison tables.

-Results: Since in the second round, both outdoor and indoor data are available, a comparison between outdoor and indoor public places could be added.

-Discussion: p. 18, 2nd paragraph: "Furthermore, mask use is declined in the non-slum region of Chennai when compared to the first round": You could also mention other comparisons between the first and the second round of the study, too, and explain or hypothesize the causes of these changes (eg. change in the number of the daily new COVID cases, new imposed COVID-19 restrictions etc).

-Discussion: you could also compare the compliance rates according to the age and the gender with other corresponding studies. For instance:

- Capraro V, Barcelo H. The effect of messaging and gender on intentions to wear a face covering to slow down COVID-19 transmission. Journal of Behavioral Economics for Policy. 2020a.

- Labiris G, Panagiotopoulou EK, Perente A, et al. Determinants of compliance to the facemask directive in Greece: A population study. PLoS One. 2021;16(3):e0248929.

- Solomou I, Constantinidou F. Prevalence and Predictors of Anxiety and Depression Symptoms during the COVID-19 Pandemic and Compliance with Precautionary Measures: Age and Sex Matter. Int J Environ Res Public Health. 2020;17(14):4924.

-References: The appropriate reference format should be followed according to the journal guidelines (Vancouver style). Some examples are the following:

-Published articles

Hou WR, Hou YL, Wu GF, Song Y, Su XL, Sun B, et al. cDNA, genomic sequence cloning and overexpression of ribosomal protein gene L9 (rpL9) of the giant panda (Ailuropoda melanoleuca). Genet Mol Res. 2011;10: 1576-1588.

- Online articles

Huynen MMTE, Martens P, Hilderlink HBM. The health impacts of globalisation: a conceptual framework. Global Health. 2005;1: 14. Available from: http://www.globalizationandhealth.com/content/1/1/14

For other examples, please refer to the guidelines of the journal.

6. PLOS authors have the option to publish the peer review history of their article (what does this mean?). If published, this will include your full peer review and any attached files.

Reviewer #1: No

Reviewer #2: No

---

## [Author Response · Author response to Decision Letter 0]

17 Jul 2021

Reviewer #1

Comment: 1. From the abstract or the introduction, this study’s primary purpose is not very clear. It would be helpful for authors to add a clear statement of purpose to the paper for ease of reading.

Response: Thank you for the comment. We have explained the purpose of the study under the section “purpose.”

Changes made: Pp-2, L: 19-25

Comment: 2. The discussion in this paper will be greatly enriched if the authors clarify the policy context. For instance, as a reader, I would like to know about the policing norm around masking and the associated fines. While authors mention it in a footnote, it should be in the main text. Also, they could explain to the reader if it is a small or significant amount by comparing it to local minimum wages. It would be helpful also to have details on the existing information campaigning around masking in Chennai.

Response: Thanks for highlighting this. We included the fine amount and compared the same. 

Changes made: P-19, L-302-306

Comment: 3. Authors did not explain how locations were chosen. A map with surveillance points will help the reader imagine this exercise more intuitively. It is okay if these locations were chosen, keeping the ease of observation in mind, but this, or any other underlying reasoning out to be stated.

Response: We plotted the study sites in the map and attached them as a figure

Changes made: Figure 1

Comment: 4. The study will also be more relevant if we can say something about masking rates in the city based on the masking rates observed in the survey. At least they comment on how the chosen surveillance points differ from the rest of the city, if at all.

Response: We ensured to give adequate representativeness for the entire Chennai city in our sample. We have done stratified sampling in the slums and non-slum areas and also provided the estimates along with 95% CI.

Changes made: Nil

Comment: 5. Is the design effect taken from prior literature? What about clustering? Further details about the sampling design would be helpful.

Response: Yes, the Design effect was calculated using the Intra Cluster Coefficient (ICCC) given in ref #9. We calculated the design effect for the results of the first round of the survey. The observed design effect was 4.5, and we used the same for sample size calculation of the subsequent rounds. 

Changes made: References #9

Comment: 6. I thought that the comparison across time periods was the most interesting. It can be complemented by information on disease spread over time: for instance, a chart that shows the evolution of the TPR over time for Chennai and the survey periods are highlighted along with making rates

Response: We have included a figure with reported cases and TPR %, and survey periods.

Changes made: Figure 2

Comment: 7. Some other comparisons are not very interesting (for instance, by Gender). The authors also don’t explain the reason for other observed differences (by region).

Response: We analysed the surveillance data by time, place, and person component. We computed the proportions by Gender and geographic regions to know the difference in behaviours and for planning the face mask campaigns. We already discussed the reasons for the differences between the regions as ‘unknown’. However, these kinds of geographical variations were observed in the USA. This was already mentioned in the manuscript

Changes made: Nil

Comment: 1) Combine Replace Tables 1 and 2 with Table 4.

Response: Thank you for the suggestion. We believe that the combined tables would be difficult 1 and 2 with 4 is difficult as the headers are different. Furthermore, readers may find it difficult to follow. We combined tables 1& 2, and 4 & 5. 

Changes made: Tables 1 & 3

Comment: 2) Clarify in table notes what values in the parenthesis mean, etc.

Response: We mentioned that the values in the parentheses denote (%) in the header of the tables

Changes made: Nil

Comment: 3) Review endnotes. For instance, endnote 7 links to a media report about fine for not masking, but it is linked to part where authors are discussing survey design.

Response: Thank you for pointing this error. We updated the reference manager links and checked. 

Changes made in the references section

Reviewer #2

Comment: Abstract: The subsection title “Introduction” could be replaced with the title “Purpose”

Response: Thank you for the suggestion. We replaced the ‘introduction’ with ‘Purpose’

Changes made: P-2 Abstract

Comment: Introduction / p. 5, 2nd paragraph: Please remove the comma before the full stop in “...settings,.”

Response: Thank you for pointing the typo. We removed the Oxford comma.

Changes made: P-5, L 88

Comment: -Methods: Please describe in more details the age classification. Do you estimate approximately the age of each individual according to his/her appearance?

Response: Yes, the data collectors recorded the age group based on the individual appearance. For the ease of observation, we categorized the age group into four categories for ease of observation. It is included in the manuscript 

Changes made: P-8, L-151-153

Comment: Results/ p. 8, last paragraph: "42% vs. 29% vs. 36%": Do you mean "42% vs. 29% for non-slums and slums, respectively" or "42% vs. 36% for non-slums and slums, respectively". Please make it clear.

Response: We apologise for the unclear statements. We updated the text as “Mask compliance was similar across three regions in the slums, however, in the non-slums, southern region (42%) had higher compliance than north (29%, p<0.001) and central region (36%, p=0.03).”

Changes made: P-9, L-177-180

Comment: Results, (eg Tables 1, 2, etc): When p>0.001), you are suggested to write the exact p value.

Response: We apologise for the error. We have corrected in all the tables as suggested. 

Changes made: Tables 1, 2, 3

Comment: Results, p. 8: “The appropriate mask use compliance did not differ between morning and evening both in the slums (29% vs. 26%; p>0·05) and non-slums (37% vs. 35%; p>0·05) areas of Chennai”: It would be very interesting if you made this comparison for the second round both for outdoor and indoor public places, too.

Response: Many thanks for the suggestion. In the first phase, we surveyed in the morning and evening. As there were no changes in the compliance between morning and evening across the study sites. Therefore, we surveyed during the morning in the second round. Unfortunately, such comparison can’t be possible. 

Changes made: Nil

Comment: -Results, p. 11: “Among the 1524 individuals who wore masks in the slums and non-slums, 1220 (80%) worn cloth masks, 278 (18%) worn medical masks, and 26 (2%) used N95 masks.”: It would be very interesting if you made this comparison for the first round, too.

Response: We did not include the type of mask variable in the first round of survey. While discussing the results with the stakeholders we realized that they were interested to know the type of mask usage. So, we included the variable in the second round of the survey.

Changes made: Nil

Comment: Additionally, the data for the type of mask (Medical /cloth/ N95 mask) could be added in the comparison tables.

Response: We have data regarding type of mask usage for the round-2 (Dec 2020) only. To avoid potential duplication. We mentioned the data in the text.

Changes made: Nil

Comment: Results: Since in the second round, both outdoor and indoor data are available, a comparison between outdoor and indoor public places could be added.

Response: Thank you for the valuable suggestion. We compared the indoor and outdoor mask compliance by slum and non-slum. 

Changes made: P-12, L- 223-226

Comment: Discussion: p. 18, 2nd paragraph: “Furthermore, mask use is declined in the non-slum region of Chennai when compared to the first round”: You could also mention other comparisons between the first and the second round of the study, too, and explain or hypothesize the causes of these changes (eg. change in the number of the daily new COVID cases, new imposed COVID-19 restrictions etc).

Response: Thank you for pointing this. We have discussed the reasons/hypothesis in the manuscript. 

Changes made: P-18-19, L-285-298

Comment: Discussion: you could also compare the compliance rates according to the age and the Gender with other corresponding studies. For instance:

- Capraro V, Barcelo H. The effect of messaging and gender on intentions to wear a face covering to slow down COVID-19 transmission. Journal of Behavioral Economics for Policy. 2020a.

- Labiris G, Panagiotopoulou EK, Perente A, et al. Determinants of compliance to the facemask directive in Greece: A population study. PLoS One. 2021;16(3):e0248929.

- Solomou I, Constantinidou F. Prevalence and Predictors of Anxiety and Depression Symptoms during the COVID-19 Pandemic and Compliance with Precautionary Measures: Age and Sex Matter. Int J Environ Res Public Health. 2020;17(14):4924.

Response: Thank you for the suggestion. We discussed and cited the suggested citations in the discussion section 

Changes made: Discussion & reference #19, #20 & #21

Response to the queries of the In-house team

Comment: 1. It appears that your ORCiD iD has not been validated in your Editorial Manager account and we are unable to proceed until that step is complete.

Response: I have validated the ORCID ID in the editorial manager portal. 

Comment: 2. We note your current Data Availability Statement is:

"No - some restrictions will apply"

"Data cannot be shared publicly because of Institutional Policy on Data sharing. However, Data are available from at request."

PLOS journals require authors to make all data necessary to replicate their study’s findings publicly available without restriction at the time of publication. When specific legal or ethical restrictions prohibit public sharing of a data set, authors must indicate how others may obtain access to the data. (https://journals.plos.org/plosone/s/data-availability)

We note that you have indicated that data from this study are available upon request. PLOS only allows data to be available upon request if there are legal or ethical restrictions on sharing data publicly.

a) If there are ethical or legal restrictions on sharing a de-identified data set, please explain them in detail (e.g., data contain potentially identifying or sensitive patient information, data are owned by a third-party organization, etc.) and who has imposed them (e.g., a Research Ethics Committee or Institutional Review Board, etc.). Please also provide non0author contact information* for a data access committee, ethics committee, or other institutional body to which data requests may be sent.

*In line with our goal of ensuring long-term data availability to all interested researchers, PLOS’ Data Policy states that authors cannot be the sole named individuals responsible for ensuring data access

Response: 

The anonymised data is being shared in the Mendeley data repository and the same mentioned under the data availability statement. You can also update the statement as “The datasets generated and analysed during the current study have been deposited in the Mendeley repository: https://data.mendeley.com/datasets/b2yzdfv9rt/1.22”

"This study was funded by the Intra-mural fund of ICMR- National Institute of Epidemiology, Chennai, India. The funders had no role in study design, data collection, and analysis, decision to publish, or preparation of the manuscript."

We have removed the funding statement from the manuscript. Please update the Amended statement as below:

“This study was supported by the Intra-mural fund of ICMR-National Institute of Epidemiology, Chennai, India. The funders had no role in study design, data collection, and analysis, decision to publish, or preparation of the manuscript.”

Comment:

We note that one or more of the authors are employed by a commercial company: Greater Chennai Corporation

Response:

The Greater Chennai Corporation (GCC) is a Government Organization. GCC is the governing body of the city of Chennai under Government of Tami Nadu. All activities of the GCC are non-profit and funded by the government of Tamil Nadu, India. The official website of GCC as follows https://chennaicorporation.gov.in/gcc/

Comment

4. Please include a copy of Tables 4 and 5 which you refer to in your text on page 14.

Response: 

We updated the tables list on the manuscript

Comment

5. Please ensure that you refer to Table 2 in your text as, if accepted, production will need this reference to link the reader to the Table.

Response

Yes, we have updated the table 2 in the manuscript section. 

Comment

6. Please provide additional details regarding participant consent. In the ethics statement in the Methods and online submission information, please ensure that you have specified (1) whether consent was informed and (2) what type you obtained (for instance, written or verbal, and if verbal, how it was documented and witnessed). If your study included minors, state whether you obtained consent from parents or guardians. If the need for consent was waived by the ethics committee, please include this information.

Response

As the study did not had any interaction with the study participants, we did not get any informed consent from the participants.

Comment

7. We note your current Data Availability Statement is:

"No - some restrictions will apply"

"Data cannot be shared publicly because of Institutional Policy on Data sharing. However, Data are available from at request."

PLOS journals require authors to make all data necessary to replicate their study’s findings publicly available without restriction at the time of publication. When specific legal or ethical restrictions prohibit public sharing of a data set, authors must indicate how others may obtain access to the data. (https://journals.plos.org/plosone/s/data-availability)

We note that you have indicated that data from this study are available upon request. PLOS only allows data to be available upon request if there are legal or ethical restrictions on sharing data publicly.

a) If there are ethical or legal restrictions on sharing a de-identified data set, please explain them in detail (e.g., data contain potentially identifying or sensitive patient information, data are owned by a third-party organization, etc.) and who has imposed them (e.g., a Research Ethics Committee or Institutional Review Board, etc.). Please also provide non0author contact information* for a data access committee, ethics committee, or other institutional body to which data requests may be sent.

*In line with our goal of ensuring long-term data availability to all interested researchers, PLOS’ Data Policy states that authors cannot be the sole named individuals responsible for ensuring data access

Response:

The anonymised data is being shared in the Mendeley data repository and the same mentioned under the data availability statement. You can also update the statement as The datasets generated and analysed during the current study have been deposited in the Mendeley repository: https://data.mendeley.com/datasets/b2yzdfv9rt/1 [22]

---

## [Decision Letter · Decision Letter 1]

9 Aug 2021

PONE-D-21-08966R1

Surveillance for face mask compliance, Chennai, Tamil Nadu, India, October-December, 2020

PLOS ONE

Dear Dr. Kaur,

Thank you for submitting your manuscript to PLOS ONE. After careful consideration, we feel that it has merit but does not fully meet PLOS ONE’s publication criteria as it currently stands. Therefore, we invite you to submit a revised version of the manuscript that addresses the points raised during the review process.

ACADEMIC EDITOR: Please address the concerns raised by the reviewer 1 and revise your submission accordingly before it can be accepted.

We look forward to receiving your revised manuscript.

Kind regards,

Amitava Mukherjee, ME, Ph.D.

Academic Editor

PLOS ONE

Journal Requirements:

Additional Editor Comments (if provided):

Reviewers' comments:

Reviewer's Responses to Questions

**Comments to the Author**

1. If the authors have adequately addressed your comments raised in a previous round of review and you feel that this manuscript is now acceptable for publication, you may indicate that here to bypass the “Comments to the Author” section, enter your conflict of interest statement in the “Confidential to Editor” section, and submit your "Accept" recommendation.

Reviewer #1: All comments have been addressed

Reviewer #2: (No Response)

2. Is the manuscript technically sound, and do the data support the conclusions?

Reviewer #1: Yes

Reviewer #2: Yes

3. Has the statistical analysis been performed appropriately and rigorously? 

Reviewer #1: Yes

Reviewer #2: Yes

4. Have the authors made all data underlying the findings in their manuscript fully available?

Reviewer #1: Yes

Reviewer #2: Yes

5. Is the manuscript presented in an intelligible fashion and written in standard English?

Reviewer #1: Yes

Reviewer #2: Yes

6. Review Comments to the Author

Reviewer #1: Referee Report

This paper reports the result of a surveillance study on masking prevalence in Chennai, Tamil Nadu, India. The authors carried out two surveillance surveys twice in Chennai in October and December 2020. The sample was chosen such that it included both slum and non-slum areas. Further, the authors conducted an additional surveillance exercise in indoor spots close to the outdoor surveillance spots.

I have added feedback to the comments that have been adequately addressed by the authors in green and suggested additional feedback in red.

Comments:

1) From the abstract or the introduction, this study's primary purpose is not very clear. It would be helpful for authors to add a clear statement of purpose to the paper for ease of reading.

Feedback: This comment has been adequately addressed by the authors.

2) The discussion in this paper will be greatly enriched if the authors clarify the policy context. For instance, as a reader, I would like to know about the policing norm around masking and the associated fines. While authors mention it in a footnote, it should be in the main text. Also, they could explain to the reader if it is a small or significant amount by comparing it to local minimum wages. It would be helpful also to have details on the existing information campaigning around masking in Chennai.

Feedback: This comment has been adequately addressed by the authors.

3) Authors did not explain how locations were chosen. A map with surveillance points will help the reader imagine this exercise more intuitively. It is okay if these locations were chosen, keeping the ease of observation in mind, but this, or any other underlying reasoning out to be stated.

Feedback: While Figure 1 is helpful, authors could still add some description on how study sites were chosen.

4) The study will also be more relevant if we can say something about masking rates in the city based on the masking rates observed in the survey. At least they comment on how the chosen surveillance points differ from the rest of the city, if at all.

Feedback: This comment has been adequately addressed by the authors.

5) Is the design effect taken from prior literature? What about clustering? Further details about the sampling design would be helpful.

Feedback: The link in footnote 9 is takes to a webpage that is unavailable.

6) I thought that the comparison across time periods was the most interesting. It can be complemented by information on disease spread over time: for instance, a chart that shows the evolution of the TPR over time for Chennai and the survey periods are highlighted along with making rates

Feedback: While Figure 2 is helpful, authors could add some discussion on what to infer from differences in masking rates given the differences in the TPR across time.

7) Some other comparisons are not very interesting (for instance, by gender). The authors also don't explain the reason for other observed differences (by region).

Feedback: This comment has been adequately addressed by the authors.

Other minor comments:

1) Combine Replace Tables 1 and 2 with Table 4.

Feedback: This comment has been adequately addressed by the authors.

2) Clarify in table notes what values in the parenthesis mean, etc.

Feedback: This comment has been adequately addressed by the authors.

3) Review endnotes. For instance, endnote 7 links to a media report about fine for not masking, but it is linked to part where authors are discussing survey design.

Feedback: This comment has been adequately addressed by the authors.

Reviewer #2: Although the numbering of lines was sometimes confusing, all comments have been addressed.

As a minor comment, authors could also change the following phrase (numbering according to the manuscript without track changes):

p 16 / Discussion / line 276: "Individuals who wear spectacles" instead of "Individuals those wear spectacles"

7. PLOS authors have the option to publish the peer review history of their article (what does this mean?). If published, this will include your full peer review and any attached files.

Reviewer #1: **Yes: **Madhulika Khanna

Reviewer #2: No

---

## [Author Response · Author response to Decision Letter 1]

11 Aug 2021

We thank the reviewers for their time and valuable feedback to improve the quality of the manuscript. Please find our responses to the feedback below. 

Response to the reviewers’ feedback

Reviewer #1 Feedback: While Figure 1 is helpful, authors could still add some description on how study sites were chosen.

Response: We mentioned the random selection of streets for the first and second round of survey under the methods section- page #6, Line 116-118 & page #7, Line 119-120. 

Reviewer #1 Feedback: The link in footnote 9 is takes to a webpage that is unavailable

Response: Thank you for pointing this issue. We assumed that you would have clicked the hyperlink of the reference #9 from the pdf to access the website. The line number 371 interrupted the web address which resulted in unavailable webpage. We have validated the link by copy pasting it in the browser and it is active. You can find this here https://preventepidemics.org/wp-content/uploads/2020/08/Promoting-Mask-Wearing-During-COVID-19.pdf

Reviewer #1 Feedback: While Figure 2 is helpful, authors could add some discussion on what to infer from differences in masking rates given the differences in the TPR across time. 

Response: We have discussed in the discussion section of the manuscript- clean version- Page #16, Line 277-282.

Reviewer #2 Feedback: p 16 / Discussion / line 276: "Individuals who wear spectacles" instead of "Individuals those wear spectacles"

Thank you for flagging this error, we have corrected in the manuscript.

---

## [Editor Report · Decision Letter 2]

9 Sep 2021

Surveillance for face mask compliance, Chennai, Tamil Nadu, India, October-December, 2020

PONE-D-21-08966R2

Dear Dr. Kaur,

We’re pleased to inform you that your manuscript has been judged scientifically suitable for publication and will be formally accepted for publication once it meets all outstanding technical requirements.

Kind regards,

Amitava Mukherjee, ME, Ph.D.

Academic Editor

PLOS ONE
---

## [Editor Report · Acceptance letter]

17 Sep 2021

PONE-D-21-08966R2 

Surveillance for face mask compliance, Chennai, Tamil Nadu, India, October-December, 2020 

Dear Dr. Kaur:

I'm pleased to inform you that your manuscript has been deemed suitable for publication in PLOS ONE. Congratulations! Your manuscript is now with our production department. 

Kind regards, 

on behalf of

Professor Dr. Amitava Mukherjee 

Academic Editor

PLOS ONE